# Impact of Patient-Centered and Self-Care Education on Diabetes Control in a Family Practice Setting in Saudi Arabia

**DOI:** 10.3390/ijerph20021109

**Published:** 2023-01-08

**Authors:** Ali I. AlHaqwi, Marwa M. Amin, Bader A. AlTulaihi, Mostafa A. Abolfotouh

**Affiliations:** 1Department of Family and Community Medicine, King Abdul-Aziz Medical City, King Saud Ben Abdu Aziz University for Health Sciences (KSAU-HS), Ministry of National Guard-Health Affairs, Riyadh 22490, Saudi Arabia; 2King Abdullah International Medical Research Center (KAIMRC), King Saud Ben Abdul Aziz University for Health Sciences (KSAU-HS), Ministry of National Guard-Health Affairs, Riyadh 11481, Saudi Arabia

**Keywords:** T2DM, glycemic, cardiovascular, HbA1c, diabetes education, diabetes management, diabetes self-management education and support (DSMES), Saudi Arabia

## Abstract

Background: Diabetes mellitus is a chronic and complex medical disease that leads to significant morbidity and mortality. Patient-centered diabetes education that emphasizes active patient involvement, self, and shared care constitutes a substantial and essential component of the comprehensive diabetes management approach. Objectives: To assess the impact of patient-centered diabetes education sessions on the prescribed treatment plan in controlling diabetes and other related cardiovascular risk factors. Methods: In a pre-experimental pretest-posttest one group study design, all referred patients with type 2 diabetes (T2DM) to the diabetes educator clinic (*n* = 130 patients) during the period of 6 months from January to July 2021 were subjected to multiple and consecutive patient-centered diabetes education sessions, based on the framework published by the Association of Diabetes Care and Education Specialties (ADCES), in addition to their usual treatment plan. Demographic, social, and biological data were obtained at the baseline, three months, and six months after the intervention. Nonparametric Friedman and Cochran’s Q tests for related samples were applied to examine the impact of this educational intervention on glycosylated hemoglobin (HbA1c) and other associated cardiovascular risks. The results of 130 patients with T2DM showed a significant reduction of mean systolic blood pressure “SBP” (*p* = 0.015), glycosylated hemoglobin (HbA1c) (*p* < 0.001), fasting blood sugar “FBS” (*p* < 0.001), total cholesterol (*p* < 0.001), low-density lipoprotein (*p* < 0.001), and triglyceride (*p* < 0.001), and significant rise of mean high-density lipoprotein (*p* = 0.011). At three and six months after the intervention, 43% and 58% of patients showed improved HbA1c levels. The mean HbA1c was reduced from 10.2% at the beginning of the study to 8.7% (*p* < 0.001) after six months. Moreover, a significant reduction in the prevalence of obesity (*p* = 0.018), high FBS (*p* = 0.011), and high SBP (*p* = 0.022) was detected. Conclusions: This study showed a considerable positive impact of diabetes education and patient-centered care on optimizing glycemic and other cardiovascular risk control. The needs of certain patients with T2DM should be addressed individually to achieve the best possible outcomes. Further research is needed to explore the long-term benefits of this intervention.

## 1. Introduction

Type 2 diabetes mellitus (T2DM) is a complex medical problem characterized by chronic hyperglycemia. It is a global concern that has a considerable negative impact on different aspects of the life of individuals and the resources of communities. Affected persons will be at significantly higher risk of life-threatening medical complications of T2DM, compromised quality of life, and increased mortality. T2DM is a leading cause of cardiovascular morbidities and mortalities, renal complications, amputations, and visual loss [1,2]. In addition, T2DM has a remarkable economic and social adverse impact due to its chronic and progressive nature [3]. International Diabetes Federation (IDF) estimated that 451 million adults were affected by diabetes in 2017 worldwide. This number is expected to be about 700 million by 2045. IDF estimated the prevalence of diabetes to be 18.5% in Saudi Arabia in 2017 [4]. These estimates included all types of Diabetes; T1DM, T2DM, and Gestational Diabetes Mellitus (GDM); however, T2DM constituted about 90% of all forms of diabetes [4]. Nationwide, studies in Saudi Arabia confirmed an increase in the prevalence of T2DM and pointed toward its alarming magnitude. The prevalence of type 2 DM rose from 23.7% in 2000 to 30% in 2011 [5,6,7]. The long-term complications of T2DM were of significant impact on the health of the Saudi population and constrained available resources to a very serious situation [1,7].

This data points to the urgent need to implement specific measures to manage T2DM effectively. Health promotion measures that focus on changing the lifestyle of individuals and reducing the predisposing risk of cardiovascular risk, including T2DM, were among the aims of Saudi health authorities [8]. These include implementing measures that emphasize the strategy of patient-centered care. Responding to patients’ needs, concerns, and expectations and encouraging healthy behaviors are the main elements of this strategy. In addition, the patient-centered approach encourages active patient involvement and shared care with the managing team at all steps of the management. Patients in the local setting of Saudi Arabia showed their willingness to receive more information about their medical illness, including its nature, progression, and available management options [9]. Moreover, the shared care approach was the preferred style of consultation among the Saudi population in a local study, indicating the readiness for self-care and to take an active role in managing their illness [10]. The new approach to diabetes care has been shifted toward empowering patients with diabetes through active collaboration with the health care team, self-care, and active enrollment in the management and inducing a significant change towards healthy behaviors [11]. Ongoing patient support and self-care education are considered the cornerstone for optimizing the care of persons with diabetes [12,13].

Standards for Diabetes Self-Management Education and Support (DSMES) have been identified to ensure the provision of comprehensive and high-quality care for patients with diabetes in different settings [12,13]. Further evaluation of these standards conformed to their scientific justifications and emphasized the importance of considering individual needs, preferences, and expectations upon implementation [14,15,16]. A sensitive and empathic communication approach is of great importance when delivering care to patients with diabetes. Providers of diabetes care should express respectful, nonjudgmental language based on patients’ strengths and achievements. The language should be free of stigma and foster collaboration between patients and providers of care [17].

The benefits of DSMES involve all aspects of the life of patients with diabetes, including clinical, psychosocial, and behavioral characteristics. The available evidence showed that DSMES reduces all-cause mortality in people with T2DM [18]. Provision of continuing DSMES is crucial, especially at critical events such as diagnosis, complications, when not meeting the treatment targets, or when transferring care [19].

Patients’ autonomy, values and preferences are of significant importance to be considered in planning and conducting patient-centered education activities. These include preventive, curative, and health promotion aspects. The consideration of some of these issues, such as using gender-responsive language to promote the utilization of immunization, was found to be effective in achieving more patients’ involvement, satisfaction and better outcomes [20].

There is limited information about the role of patient-centered and self-care education in the management of diabetes in the local settings of Saudi Arabia. This study aimed to assess the impact of patient-centered diabetes education sessions on the prescribed treatment plan in controlling diabetes and other related cardiovascular risk factors.

## 2. Methods

### 2.1. Study Design

This is a pre-experimental pretest-posttest one group study design, where the patients were assessed three times: at baseline, three months, and six months after the intervention. The study was conducted at Khashm Al Aan, a family practice clinic at King Abdul-Aziz Medical City, National Guard Health Affairs. This well-staffed major family practice center provides comprehensive medical services for National Guard employees and their dependents. These services include the management of acute and chronic medical diseases, including diabetes.

Diabetes care is provided by a team of board-certified family physicians, nurses, pharmacists, diabetes educators, and other support staff. Diabetes education clinics offer health education services for newly diagnosed and known patients with diabetes. These clinics run daily services and can accept patients with diabetes on a walk-in basis and by appointment.

### 2.2. Study Participants

All adult male and female patients with type 2 diabetes referred to the diabetes educator clinics were included during the period of six months from January to July 2021. Patients with mental impairment or diseases that could affect proper communication and judgment were excluded.

### 2.3. Education Intervention

American Association of Diabetes Educators (AADE) recommended a framework for effective diabetes self-management [11]. Recently, the AADE became the Association of Diabetes Care and Educators Specialties (ADCES) [16]. The proposed framework ADCES7 aims to cover seven essential factors required to achieve better outcomes in preventing acute and chronic complications of diabetes and other cardiometabolic conditions. These factors cover areas related to healthy eating, physical activity, monitoring of blood glucose, adherence to medications, problem-solving skills, healthy coping for a better quality of life, and reducing risks. Problem-solving skills are highly needed for patients with diabetes to enable them to handle acute and chronic complications of the disease or its medications, such as hypoglycemia and hyperglycemia. As some of these problems might be life-threatening, a systematic approach is taught to patients during these sessions for early anticipation and prompt intervention. Health coping strategies include teaching patients’ strategies to fulfill health care and psychosocial obligations for diabetes care as healthy diet alternatives, expressing emotions, and incorporating physical activities in the daily life of patients. Evidence showed that complications of diabetes were fourfold higher among patients with diabetes who had not received diabetes self-care educational sessions compared to those who received and were involved in active self-care formal educational activities [14,15,16,18]. Better glycemic control was reported with individuals engaged in self-management diabetic education [21].

All patients with T2DM referred to the diabetes educator clinics were included during the period of the study. The needs of these patients were assessed, and an initial 30–45 min patient-centered educational session was given to all of them based on the ADCES7 framework. The diabetes educator reviewed the medical records of all referred patients, obtained available biological and laboratory data for better assessment, and shared them with patients. These data were gathered and documented on a “data sheet” designed to meet the purpose of this study. General patients’ diabetes knowledge and skills were assessed by checking whether the patient had received any form of diabetes education. Needs for glucose home monitoring were also reviewed, including checking the availability of a glucometer and providing one if needed. Then, patients and educators determine patients’ needs and identify priorities of the knowledge, skills, and behaviors in the ADCES7 framework. Finally, patients and care providers agree upon the clinical outcomes that will be monitored and checked in further visits.

ADCES has defined the ADCES7 Self-Care behaviors as essential for effective diabetes self-management. These behaviors are; healthy eating, being active, monitoring, taking medications, problem-solving, healthy coping, and reducing risks [16].

The initial diabetes education session was conducted with patients individually and in the presence of other family members if needed or requested by patients. Illustrations and materials were utilized for better knowledge and skills acquisition. Educational brochures on important topics were provided to participants as required. Examples of these educational materials include topics on features and management of hyperglycemia and hypoglycemia, insulin therapy, Diabetes and exercise, and diet and Diabetes. All patients were given a glucose monitoring sheet to document fasting and postprandial home glucose readings. Patients were informed about the targets of each task and how to achieve them. They were informed as well that the educator is accessible to discuss any concern or needs related to glucose monitoring and readjustment of medications. Patients were instructed to contact the educator by phone or come as a walk-in.

The ADCE7 Self-Care behaviors were reviewed again with patients at their follow-up visits, at three and six months, along with gathering other biological and laboratory data. In addition, the plan of treatment and achieved targets were reviewed at each visit by patients, treating physicians, and diabetes educators.

The education intervention was planned to be patient-centered, acceptable to patients, and based on their preferences, needs, and expectations. For instance, if an initial review of the educator showed that the patient was started on insulin recently for better glycemic control, the concerns of patients would be explored, and needs addressed mutually. In many instances in this example, specific tasks of the ADCES7, taking medication, monitoring, and problem-solving, especially the occurrence of hypoglycemia, were emphasized and highlighted. In addition, as in patients with abnormal BMI and sedentary lifestyle, priority was given to eating and activity areas if both educator and patients thought these were a priority.

Every patient involved in this study received about 10 h of diabetes education and support for the study. Patients who did not reach the desired clinical or biological outcomes or showed deterioration at three and six months from the beginning of the study were reported to the treating physician to take the necessary and appropriate actions. These actions included review and adjustment of medications. Referral to a specialized diabetic center was considered as well.

The diabetes education in this study was provided by one of the authors, MMA. She is a registered nurse who has been running the diabetes education clinic since 2012. She attended many local diabetes education courses as well as an advanced postgraduate course conducted by the American Diabetes Association. She is currently doing a postgraduate diploma in Diabetes Care at the University of Warwick, UK.

### 2.4. Sample Size and Sampling Technique

To detect a difference of at least 1% in the level of Glycosylated Hemoglobin (HbA1c) between two readings before and after the intervention at a 5% level of significance with a standard deviation of 2.5 and to achieve a power of at least 80%, the required sample size is 70. To accommodate for a 20% dropout rate at three and six months, the total sample size was around 100 patients. The calculation was done using the Piface sample size application [22]. A total of 130 patients were subjected to the education intervention. Non-probability convenient sampling was used by including patients who met the inclusion criteria until the sample size was met. All patients complied with the intervention till the end of 6 months.

## 3. Data Collection

Data were collected on patients’ demographical characteristics (age, gender, education, etc.), presence of comorbidities, clinical indicators, and laboratory blood levels of HbA1c, Low-Density Lipoprotein (LDL), High-Density Lipoprotein (HDL), Total Cholesterol (TC), Triglycerides (TG), and Fasting Blood Sugar (FBS). This data was measured at baseline, three months, and six months. The primary outcome variable to indicate glycemic control is HbA1c.

### 3.1. Operational Definitions

For this study, target levels for clinical and laboratory variables were as follows: body mass index “BMI” (less than 25 kg/m^2^), Blood pressure (Systolic blood pressure “SBP” less than 140 mmHg and diastolic blood pressure “DBP” less than 90 mmHg), Total cholesterol “TC” (less than 5.1 mmol/L), low-density lipoprotein “LDL” (less than 3.37 mmol/L), high-density lipoprotein “HDL” (more than 1.5 mmol/L), triglycerides “TG” (less than 2.3 mmol/L), Hemoglobin A1c “HbA1c” (7%) and fasting blood sugar “FBS” (≤7 mmol/L) [23]. The prevalence of abnormal parameters was calculated as the percentage of patients with values above (or below, as in the case of HDL levels) these target values. Furthermore, patients were categorized into the following 4 categories according to the level of HbA1c: ≤7%, 7.1–9%, 9.1–11%, and >11% [24], and improvement after the intervention was considered if the patient’s baseline category changed to another category with lower values, while deterioration was considered if his/her baseline category changed to another category with higher values.

### 3.2. Ethics Approval and Consent to Participate

This research was approved by the Institutional Review Board (IRB) of The Ministry of National Guard-Health Affairs, Riyadh, Saudi Arabia (Ref. RC20/623/R). Participation in this study was voluntary. Patients were assured in written informed consent that their data would remain anonymous and that their participation would not affect their current health services. The purpose of the study was explained, and all issues regarding confidentiality and privacy were assured and protected at all times.

### 3.3. Data Analysis

Data were analyzed using the SPSS statistical software version 24(IBM, Armonk, NY, USA). The data were checked for normality of HbA1c levels and cardiovascular risk factors before deciding which statistical test to use using the Shapiro–Wilk test. Categorical variables were described using frequency and percentages. Mean and Standard Deviation (SD) were used for continuous variables, such as age and laboratory results if normally distributed; otherwise, median and interquartile range (IQR) were used. The Friedman test, a non-parametric statistical test similar to the parametric repeated measures ANOVA, was used to detect differences in the impact of intervention across multiple test attempts, and Cochran’s Q test for related samples, a non-parametric statistical test was applied to test the change of prevalence of different clinical and laboratory parameters three and six months after education intervention. Significance was considered at *p*-value < 0.05.

## 4. Results

A total of 130 participants were involved in this study with a mean (SD) age of 58 (8.1) years. The majority of patients were females (89.2%) and married (81.5%). Diabetes was less than five years in 43% of the sample, whereas for 25.2% and 31.5% of participants, the durations were 5–10 years and >10 years, respectively. Most of the participants were on different combinations of pharmacological agents. Metformin was used by 86.9% and other oral hypoglycemics by 75.4% of the sample. Of all participants, 71.5% and 9.2% used insulin and Glucagon-like peptide 1, respectively, Table 1.

Table 2 shows changes in median (IQR) values of HbA1c and other cardiovascular risk parameters before and after education intervention among patients with diabetes. Friedman test for related samples was applied to test the trend of clinical and laboratory indicators before and 3, and 6 months after education intervention. The results showed a significant reduction of SBP (χ^2^ = 8.33, *p* = 0.015) and HbA1c (χ^2^ = 97.57, *p* < 0.001), and FBS (χ^2^ = 71.60, *p* < 0.001), Figure 1. At 3 months, there was a significant reduction of HbA1c (*p* < 0.001) and FBS (*p* < 0.001). At six months, the significant decreases for HbA1c (*p* < 0.001) and FBS (*p* < 0.001) were retained, in addition to a substantial reduction in SBP (*p* < 0.022). From three to six months, significant improvement of HbA1c (*p* = 0.001) and FBS (*p* = 0.008) occurred. HbA1c levels showed a median (IQR) level of 10.3% (9.2–11.1%) at the baseline, which was reduced to 9.2% (8.0–10.1%) and 8.6% (7.8–9.7%) at three months and six months of the intervention, respectively. As shown in Figure 1, the median level of HbA1c was reduced by 1.5% at 6 months of intervention.

Figure 2 shows the laboratory indicators of participants at the baseline, three months, and six months of the intervention using the Friedman test for related samples. The results showed a significant reduction of HbA1c (χ^2^ = 97.57, *p* < 0.001), TC (χ^2^ = 25.87, *p* < 0.001), LDL (χ^2^ = 31.07, *p* < 0.001), TG (χ^2^ = 16.44, *p* < 0.001), and significant rise of HDL (χ^2^ = 9.10, *p* = 0.011). At 3 months, there was a significant reduction of TC (*p* < 0.001), LDL (*p* < 0.001), and TG (*p* = 0.009). At 6 months, significant reductions were retained for TC (*p* < 0.001), LDL (*p* < 0.001), and TG (*p* < 0.001). From three months to six months, significant improvement of only TG (*p* = 0.022) occurred, Table 2.

Cochran’s Q test for related samples was applied to test the change of prevalence of different clinical and laboratory parameters three and six months after education intervention. Figure 3 shows the significant reduction in the prevalence of abnormal HbA1c (*p* = 0.012), obesity (*p* = 0.018), and abnormal FBS (*p* = 0.011). However, there was no significant reduction in any of the prevalence of abnormal BP (*p* = 0.35), abnormal LDL (*p* = 0.23), abnormal HDL (*p* = 0.30), abnormal TC (*p* = 0.14), and abnormal TG (*p* = 0.20).

Figure 4 shows the impact of an intervention on HbA1c levels at three and six months. Overall, 43.1% and 58.5% of patients showed improvement in HbA1c levels three and six months after intervention. The figure shows that 4.6% of participants experienced deterioration in their glycemic control at three and six months of the study compared to their baseline figures.

## 5. Discussion

The main feature of this patient-centered educational program, which has been proposed by the Association of Diabetes Care and Educators Specialties (ADCES), is to enable patients to acquire lifelong behaviors and skills [16]. These skills include managing diabetes and other risks by adopting a healthy lifestyle, developing problem-solving skills, active involvement, and acquiring coping strategies. The benefits will extend beyond glycemic control to involve better quality of life. This study aimed to examine the effect of structured diabetes education sessions, which is based on more patients’ empowerment and shared care, on glycemic control and lowering cardiovascular risks. It showed that adding these patient-centered diabetes education sessions to the prescribed treatment plan of diabetes management was effective in achieving better glycemic control and lowering other cardiovascular risks six months after this intervention. The mean HbA1c was reduced by 1.5%. Evidence from prospective studies demonstrated that a reduction of HbA1c is associated with a significant decrease in all Diabetes related complications. These include minimizing the risks of micro-vascular and macro-vascular amputations and death [25,26,27].

It has been reported that every 1% reduction in the mean level of HbA1c is associated with a 14% risk reduction of myocardial infarction, a 21% reduction in all Diabetes-related deaths, and a 37% and 43% reduction in microvascular complications and amputation risks, respectively [28]. Many studies and reviews have indicated that the optimal level of HbA1c, which should be aimed at, is 7% [29,30]. In this study, the number of participants with optimal HbA1c levels was almost doubled, as it was raised from 6.9% (93.1% abnormal HbA1c levels) at the baseline to 13.1% (86.1% abnormal HbA1c levels) at three months and this figure was maintained at six months of intervention (Figure 3). This demonstrated the benefits of this approach both in short and intermediate terms. At 6 months of this education intervention, a significant reduction in the prevalence of abnormal HbA1c (*p* = 0.0.012) was noticed, and the mean HbA1c was reduced by 1.5%. This finding was similar to the result of a previous study in a similar setting in Kuwait [31], where patients that received diabetes self-management education (DSME) sessions demonstrated better diabetes control with an average reduction of 1.3% HbA1c over 12 months compared to an average HbA1c increase of 1.1% in the control group (*p* < 0.001). Meanwhile, 43.1% and 58.5% of patients showed improvement in HbA1c levels at three and six months after intervention (Figure 4). However, it is necessary to note that improvement does not mean reducing HbA1c levels to normal levels but only reducing them to lower levels. Generally speaking, in our study, the number of patients with optimal glycemic control was almost doubled, in addition to the other reported metabolic benefits.

Many attributes that can influence glycemic control include; the age of patients, social status and support, presence of comorbidities, and psychological status [32,33]. Lack of Self-monitoring blood glucose, presence of comorbidities, duration of diabetes mellitus, physical activity of three or less than three days, total cholesterol of 200 mg/dL or more, a waist -to -hip ratio of 0.9 or greater for males and 0.85 or greater for female, and types of antidiabetic medication were the independent predictors of poor glycemic control [33]. Efforts should be made towards monitoring and controlling these factors by the concerned parties. In our study, patients and educators determined patients’ needs and identified priorities of the knowledge, skills, and behaviors in the ADCES7 framework. Patients and care providers agreed upon the clinical outcomes that would be monitored and checked in further visits. Successful diabetes self-management requires considerable knowledge on the part of patients and families, including an understanding of the effects of diabetes on the body, the goals of treatment, and the effects of various behaviors on glucose regulation [34].

There were benefits from this study, in addition to better glycemic control. A significant reduction in other cardiovascular disease (CVD) risks was noted. Randomized controlled trials demonstrated that lowering LDL reduces the CVD risks by up to 43%, and reducing systolic blood pressure was associated with up to 60% reduction in CVD risks in patients with Diabetes [35]. Diabetes care based on a multidisciplinary approach, which includes diabetes educators, is effective in achieving better glycemic and lowering cardiovascular risks in local settings [36].

Features and elements that contribute to the effectiveness of different education interventions among patients with T2DM were previously assessed by a meta-analysis [34]. These include; types of intervention, teaching methods, format, duration, and the number of contact hours [37]. Such interventions reported improvement in knowledge, behaviors, and metabolic outcomes. The effectiveness of the individual education method used by this study was supported by a randomized controlled trial, which confirmed the superiority of this method in achieving better glycemic control compared to the group education and usual care methods [38], where the mean HbA1c concentration decreased in both individual and group education, but significantly more with the individual (−0.51%) than group education (−0.27%), (*p* = 0.01). Our study showed that a considerable number of participants remained in the same glycemic control category, and about 5% even showed deterioration in their glycemic control. Such patients need further attention and monitoring of their glycemic and other cardiovascular risks. An individualized plan may be considered for high-risk patients and those who showed unexpectedly unfavorable responses or even deterioration in their glycemic or general well-being [39].

It is of remarkable importance to consider the emotions, motives, and situations of both educators and participants in order to achieve the best outcomes from patient-centered diabetes education activities [40]. Reflexivity is a considerable feature of a diabetes educator, where one’s own beliefs, judgments and practices are to be examined during the education sessions and how these may influence the process. Another aspect being considered is the awareness of the positionality of the educator and the potential effect of personal characteristics and perspectives in the design, patients’ selection and process of the educational activities [41]. Managing pre-understanding using appropriate reflection and primarily reduction is one of the measures to be utilized for negotiating positionality [42].

### Strengths and Limitations

There is limited information about the role of patient-centered and self-care education in the management of diabetes in the local settings of Saudi Arabia. Showing evidence about the effectiveness of this approach will significantly contribute to the delivery of cost-effective management and decrease the huge health and economic burden of diabetes as a major chronic disease in the Saudi community. This study is a useful pilot study to test educational intervention’s processes and potential effects. However, this single-center prospective study has some limitations. These include the small number of patients enrolled that did not allow for subgroup analyses to compare between patient groups with regard to the impact of educational intervention; limited generalizability beyond patients affiliated with this single-center, short follow-up; and lack of a control group to minimize the threats of internal validity, such as; history, maturation, and regression. Many of the possible confounding factors, such as age, gender, level of education, type of treatment, etc., were not adjusted for a while assessing the impact of educational intervention. A long-scale randomized controlled trial is recommended. A larger randomized controlled trial in areas including recruitment, intervention delivery, and follow-up.

## 6. Conclusions

This study demonstrated the positive effect of patient-centered diabetes education reflected by better glycemic and other cardiovascular risk control. Different attributes that affect patients’ adherence and outcomes should be addressed and monitored. It has been shown by this study as well that certain groups of patients with diabetes should be addressed individually to achieve the best possible outcomes. Well-organized diabetes care based on patients’ active role and shared care will significantly contribute towards minimizing the burden of diabetes. Future multicenter research is needed to explore the long-term benefits of this intervention. Qualitative studies are highly recommended for the exploration of the role of psychosocial factors and their impact on patient outcomes and well-being.

Decision-makers in Saudi Arabia should utilize the available published data on the vital role of diabetes self-management education in alleviating the burden of diabetes and achieving better clinical and economic benefits.

## Figures and Tables

**Figure 1 ijerph-20-01109-f001:**
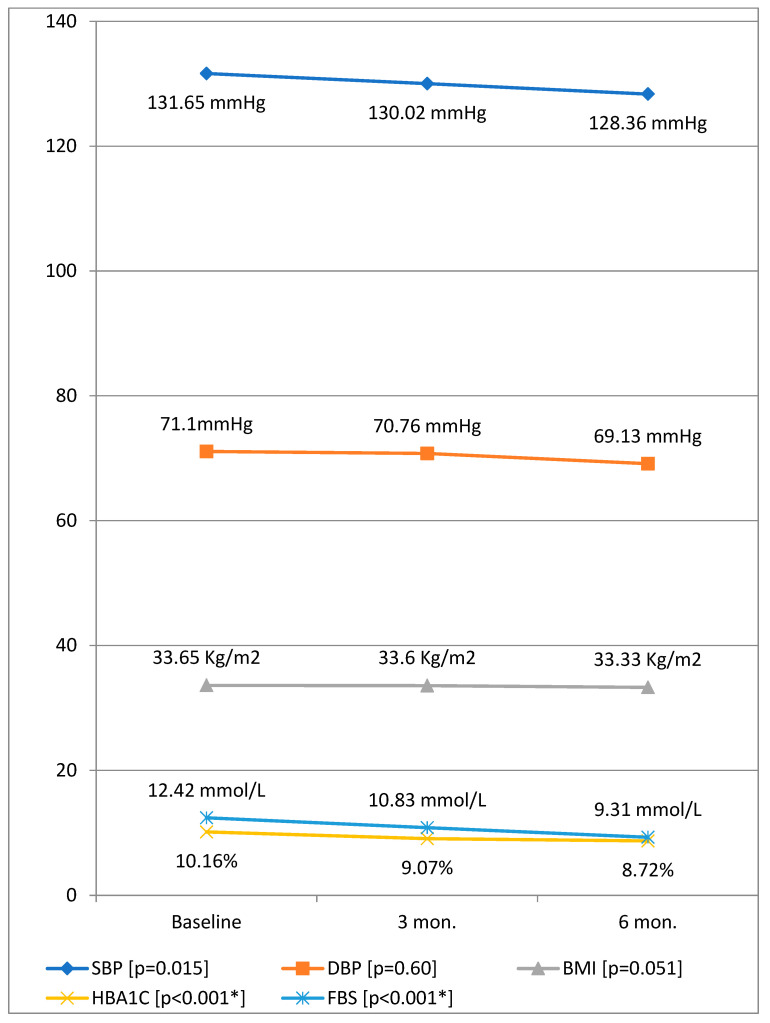
Mean values of HbA1c and other clinical parameters before and after education intervention among participants. Note: BMI—body mass index, SBP—Systolic blood pressure, DBP—diastolic blood pressure, HBP—High blood pressure, TC—Total cholesterol, LDL—low density lipoprotein, HDL—high density lipoprotein, TG—triglycerides, HbA1c—Hemoglobin A1c, FBS—fasting blood sugar, *—statistically significant.

**Figure 2 ijerph-20-01109-f002:**
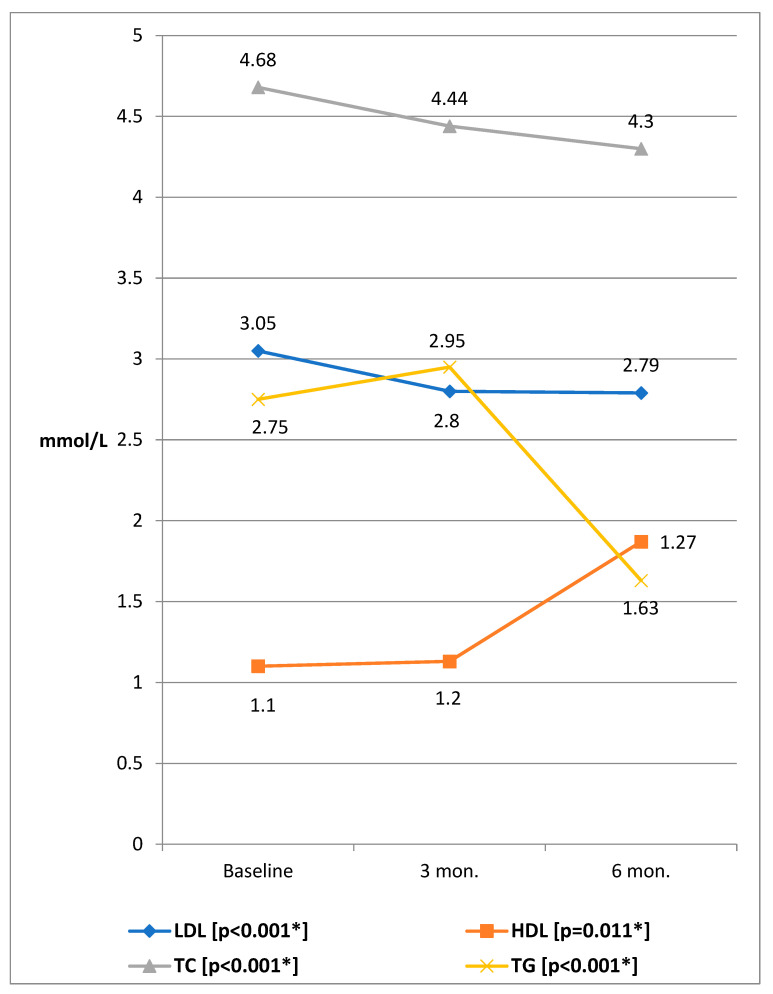
Mean values of lipid profile parameters before and after the educational intervention. Note: BMI—body mass index, SBP—Systolic blood pressure, DBP—diastolic blood pressure, HBP—High blood pressure, TC—Total cholesterol, LDL—low density lipoprotein, HDL—high-density lipoprotein, TG—triglycerides, HbA1c—Hemoglobin A1c, FBS—fasting blood sugar, *—statistically significant. *—Friedman test for related samples was applied.

**Figure 3 ijerph-20-01109-f003:**
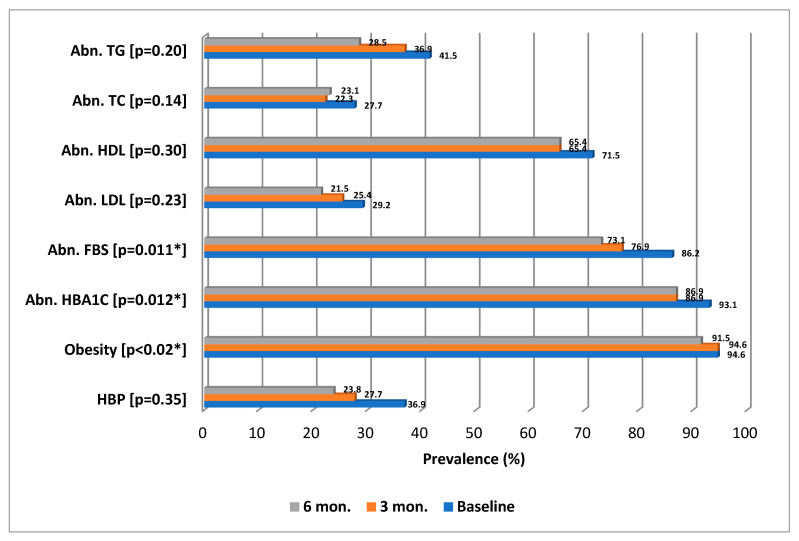
Prevalence of abnormal clinical and laboratory parameters before and after the educational intervention. Note: BMI—body mass index, SBP—Systolic blood pressure, DBP—diastolic blood pressure, HBP—High blood pressure, TC—Total cholesterol, LDL—low density lipoprotein, HDL—high-density lipoprotein, TG—triglycerides, HbA1c—Hemoglobin A1c, FBS—fasting blood sugar, *—statistically significant. *—Cochran’s Q test was applied.

**Figure 4 ijerph-20-01109-f004:**
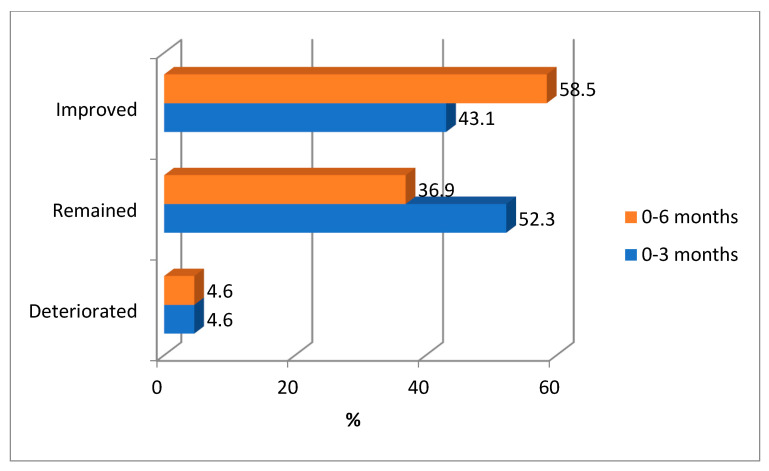
Impact of an education intervention on HbA1c levels at 3 and 6 months of intervention.

**Table 1 ijerph-20-01109-t001:** Personal and sociodemographic characteristics of participants.

Personal Characteristics	No	%
Gender		
Male	14	10.8
Female	116	89.2
Age (years)		
<50	18	13.8
50–59	61	46.9
60+	51	39.2
Education		
Educated	51	39.2
Non-educated	55	60.8
Duration of diabetes (years)		
<5	56	43.1
5–10	33	25.4
11+	41	31.5
Marital status		
Single	24	18.5
Married	106	81.5
Comorbidities		
HTN	130	100.0
DLP	117	90.0
Thyroid disease	53	40.8
Psychiatric problem	18	13.8
Bronchial asthma	22	16.9
Type of Treatment		
oral glycemic	98	75.4
metformin	113	86.9
Glucagon-like peptide 1	12	9.2
Insulin	93	71.5

HTN—Hypertension, DLP—Dyslipidemia.

**Table 2 ijerph-20-01109-t002:** Change of median (IQR) values of HbA1c and other cardiovascular risk parameters before and after education intervention among participants.

Parameter	Baseline	Three Months	Six Months	*p*–Value [0–3 Months]	*p* Value [0–6 Months]
	Median (IQR)				
SBP mmHg	130 (124–140)	130 (120–140)	130 (120–137)	0.19	0.02 *
DBP mmHg	70 (66–75)	70 (70–75)	70 (67–73)	0.92	0.14
BMI kg/m^2^	35 (30–35)	35 (30–35)	34 (30–35)	0.47	0.09
HbA1c (%)	10.3 (9.2–11.1)	9.2 (8.0–10.1)	8.6 (7.8–9.7)	<0.001 *	<0.001 *
FBS mmol/L	12.1 (8.8–15.1)	9.8 (7.3–12.0)	9.0 (7.0–11.0)	<0.001 *	<0.001 *
LDL mmol/L	2.9 (2.4–3.5)	2.6 (2.2–3.4)	2.7 (2.0–3.3)	<0.001 *	<0.001 *
HDL mmol/L	1.1 (0.9–1.3)	1.2 (0.9–1.3)	1.1 (1.0–1.3)	0.36	0.06
TC mmol/L	4.5 (4.0–5.3)	4.1 (3.7–5.0)	4.1 (3.5–5.0)	<0.001 *	<0.001 *
TG mmol/L	1.6 (1.2–2.0)	1.4 (1.0–2.0)	1.3 (1.0–1.9)	0.009 *	<0.001 *

*—Statistical significance, IQR—interquartile range, BMI—body mass index, SBP—Systolic blood pressure, DBP—diastolic blood pressure, HBP—High blood pressure, TC—Total cholesterol, LDL—low -density lipoprotein, HDL—high -density lipoprotein, TG—triglycerides, HbA1c—Hemoglobin A1c, FBS—fasting blood sugar.

## Data Availability

Most of the data supporting our findings are contained within the manuscript, and all others, excluding identifying/confidential patient data, will be shared upon request by contacting the corresponding author [Mostafa Abolfotouh mabolfotouh@gmail.com].

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
