# Peer review of "Impact of Patient-Centered and Self-Care Education on Diabetes Control in a Family Practice Setting in Saudi Arabia"

_ijerph, 2023, doi:10.3390/ijerph20021109_

Round 1
Reviewer 1 Report
Thank you for choosing me as a reviewer for this manuscript entitled ‘Impact of Patient-Centered and Self-Care Education on Diabetes Control in a Family Practice Setting’. The manuscript is befitting the vision and scope of the International Journal of Environmental Research and Public Health and addresses a very important topic of Patient-Centered Self-Care Education in a Saudi Arabian Clinic.
Overall, the manuscript is coherent and has all the key stipulated sections Abstract, Introduction, Methods, Discussion, Limitation and Conclusion. Some minor edits are suggested below, for being considered for publication.
Title: Minor tweak is suggested: Please add ‘….in Saudi Arabia, after Setting
Keywords: Please expand T2DM, and HbA1c, and add Saudi Arabia.
Introduction: Please add a few examples of patient-centered self-care education in the management of diabetes or other chronic health issues, exemplifying that they have proven to have positive health outcomes. This will strengthen the rationale of the study. A few useful citations are here:
Dutta, T., Agley, J., Lin, H. C., & Xiao, Y. (2021, May). Gender-responsive language in the National Policy Guidelines for Immunization in Kenya and changes in prevalence of tetanus vaccination among women, 2008–09 to 2014: A mixed methods study. In Women's Studies International Forum (Vol. 86, p. 102476). Pergamon.
Results and Discussion: Overall, both the Results and the Discussion sections need to be enriched and, critically appraised in the context of Saudi Arabian healthcare setting. One section explaining the core trainer’s and researcher’s reflexivity and positionality issue should be highlighted.
Please spellcheck the manuscript and revise typos such as 'trails' for 'trials' and such likes.
Author Response
Reviewer 1
Thank you for choosing me as a reviewer for this manuscript entitled ‘Impact of Patient-Centered and Self-Care Education on Diabetes Control in a Family Practice Setting’. The manuscript is befitting the vision and scope of the International Journal of Environmental Research and Public Health and addresses a very important topic of Patient-Centered Self-Care Education in a Saudi Arabian Clinic.
Overall, the manuscript is coherent and has all the key stipulated sections Abstract, Introduction, Methods, Discussion, Limitation and Conclusion. Some minor edits are suggested below, for being considered for publication.
Title: Minor tweak is suggested: Please add ‘….in Saudi Arabia, after Setting
Added to the title. Thank you. [line 3]
Keywords: Please expand T2DM, and HbA1c, and add Saudi Arabia.
Added. Thank you.[line 40]
Introduction: Please add a few examples of patient-centered self-care education in the management of diabetes or other chronic health issues, exemplifying that they have proven to have positive health outcomes. This will strengthen the rationale of the study. A few useful citations are here:
Dutta, T., Agley, J., Lin, H. C., & Xiao, Y. (2021, May). Gender-responsive language in the National Policy Guidelines for Immunization in Kenya and changes in the prevalence of tetanus vaccination among women, 2008–09 to 2014: A mixed methods study. In Women's Studies International Forum (Vol. 86, p. 102476). Pergamon.
Elaboration on patient-centered education was made.
This paragraph was added in the introduction section to highlight this point. “ Patients' autonomy, values, and preferences are of significant importance to be considered in planning and conducting patient-centered education activities. These include preventive, curative, and health promotion aspects. The consideration of some of these issues, such as using gender-responsive language to promote utilization of immunization, was found to be effective in achieving more patients' involvement, satisfaction, and better outcomes”20 [lines 89-93]
A reference was cited to support this point.[Ref.#20]
Results and Discussion: Overall, both the Results and the Discussion sections need to be enriched and, critically appraised in the context of Saudi Arabian healthcare setting. One section explaining the core trainer’s and researcher’s reflexivity and positionality issue should be highlighted.
The importance of the educator’s positionality and reflexivity was described.
This paragraph was added to the discussion: “It is of great importance to consider the emotions, motives, and situations of both educators and participants to achieve the best outcomes from patient-centered diabetes education activities.40 Reflexivity is a notable feature of diabetes educators where one's own beliefs, judgments, and practices are examined during the education sessions and how these may influence the process. Another aspect being considered is the awareness of the positionality of the educator and the potential effect of personal characteristics and perspectives in the design, patients’ selection, and process of the educational activities.41 Managing pre-understanding using appropriate reflection and primarily reduction are one of the measures to be utilized for negotiating positionality.42” [lines 393-401]
Three references were cited to support this point. [Ref.#40-42]
Please spellcheck the manuscript and revise typos such as 'trails' for 'trials' and such likes.
All spelling mistakes and grammatical errors were corrected.
Reviewer 2 Report
Overall this is an interesting study which adds to the literature. It may have important implications for Saudi Arabia or the region.
I hope my comments (not in a particular order) will be useful:
· Remove “Significance was considered at p<0.05.” from the abstract. It should be in the methods section.
· Place “The diabetes education in this study was provided by one of the authors, MMA….” in Appendix.
· Reformulate the last two statements in the introduction by clearly stating the objectives of the study.
· Remove hyperlinks from the text.
· The first statement in Section 3 should be changed / corrected “Data collection sheet was used to; patients’ demographical characteristic”.
· It would be interesting to report differences from 3 to 6 months.
· Elaborate on the clinical significance of the results presented.
· Elaborate on the mechanism behind your results. What do you think drive the results? Are there any specific groups of patients (e.g. age group) who are more / less likely to benefit from the program? I think it will be interesting to perform a subgroup analysis.
· Elaborate on the strengths of the research designed utilised.
· Limitations of this study are not well described, including specific biases the research design utilised is not able to address.
· There are multiple grammar / style mistakes in the text e.g. “Future multicenter researches…”
· The Conclusion Section would benefit from clearer implications of findings for Saudi Arabia or the region.
· In Conclusion Section “considerable positive effect” is confusing, kindly change / rephrase.
Author Response
Reviewer 2
Overall this is an interesting study which adds to the literature. It may have important implications for Saudi Arabia or the region.
I hope my comments (not in a particular order) will be useful:
- Remove “Significance was considered at p<0.05.” from the abstract. It should be in the methods section.
Removed.
- Place “The diabetes education in this study was provided by one of the authors, MMA….” in Appendix.
Done.
- Reformulate the last two statements in the introduction by clearly stating the objectives of the study.
Done.
- Remove hyperlinks from the text.
Done.
- The first statement in Section 3 should be changed / corrected “Data collection sheet was used to; patients’ demographical characteristic”.
The statement was corrected.[line 190]
- Elaborate on the mechanism behind your results. What do you think drive the results? Are there any specific groups of patients (e.g. age group) who are more / less likely to benefit from the program? I think it will be interesting to perform a subgroup analysis.
Agree. However, unfortunately, the sample size was small and did not allow for this subgroup analysis. This issue was highlighted as one of the study’s limitations in the last paragraph of the discussion section.
- Elaborate on the strengths of the research designed utilised.
Done.
- Limitations of this study are not well described, including specific biases the research design utilised is not able to address.
Added. [lines 403-414]
- There are multiple grammar / style mistakes in the text e.g. “Future multicenter researches…”
All spelling mistakes and grammatical errors were corrected
- The Conclusion Section would benefit from clearer implications of findings for Saudi Arabia or the region. In Conclusion Section “considerable positive effect” is confusing, kindly change / rephrase.
This statement was rephrased.
“This study demonstrated the positive effect of patient-centered diabetes education reflected by better glycemic and other cardiovascular risks control.” [lines 419-420]
Reviewer 3 Report
Very extensive thorough work, I would consider it important to create patient groups for the incoming criteria, to display patients treated with metformin and other tablet antidiabetics or insulin.
Author Response
Very extensive thorough work, I would consider it important to create patient groups for the incoming criteria, to display patients treated with metformin and other tablet antidiabetics or insulin.
Agree. However, unfortunately, the sample size was small and did not allow for this subgroup analysis. This issue was highlighted as one of the study’s limitations in the last paragraph of the discussion section.
Reviewer 4 Report
In general, the contribution from this paper in its current form is very marginal, thus not significant and sufficient to be considered as a full length research paper.
The Data Analysis (subsection 3.3) and Results (Section 4) need significant improvements. The statistical criteria, techniques and discussions that are used in this study are very trivial. These basic statistical measures are used without justifications. Therefore, more advanced techniques are needed. For example,
- Why the interquartile range (IQR) is used in your study?
- Is 95% Confidence Interval better than the IQR for your study?
- A comparison study between these two statistical measures is needed.
- What do you mean by Median (IQR)? Do you mean the second quartile Q2?
- Why "χ2" test is used in your study? Some discussions are needed.
- Since you have parameters before and after education, can paired samples t-test be used? Some discussions are needed.
- The figures and their related discussions are very trivial. I suggest the authors to use more advanced statistical techniques and discussions.
Author Response
Reviewer 4
n general, the contribution from this paper in its current form is very marginal, thus not significant and sufficient to be considered as a full length research paper.
The Data Analysis (subsection 3.3) and Results (Section 4) need significant improvements. The statistical criteria, techniques and discussions that are used in this study are very trivial. These basic statistical measures are used without justifications. Therefore, more advanced techniques are needed. For example,
- Why the interquartile range (IQR) is used in your study?
- Is 95% Confidence Interval better than the IQR for your study?
- A comparison study between these two statistical measures is needed.
- What do you mean by Median (IQR)? Do you mean the second quartile Q2?
- Why "χ2" test is used in your study? Some discussions are needed.
- Since you have parameters before and after education, can paired samples t-test be used? Some discussions are needed.
- The figures and their related discussions are very trivial. I suggest the authors to use more advanced statistical techniques and discussions.
All these points and queries were responded to in the data analysis section with a more detailed description of the statistical tests used and the justifications for their use. [lines 219-228, 276, 291-292]
Reviewer 5 Report
Manuscript ID: ijerph-2113521
Type of manuscript: Article
Authors: Ali I AlHaqwi, Marwa M Amin, Bader A AlTulaihi, Mostafa A.
Abolfotouh *
Reviewers General Comments
Thank you for giving me the opportunity to review this interesting and original article entitled “Impact of Patient-Centered and Self-Care Education on Diabetes Control in a Family Practice Setting”. The topic of this article is of great significance as it addresses the issue of an education gap around diabetes control. The authors rightfully highlighted that diabetes (mainly type II, T2DM) is of global concern, with an expected 700 million people who might be affected by the year 2045! Thus, there is an urgent need to implement effective care management of T2DM. There has been growing evidence that empowering patients to be more involved in their own health care contributes to better outcomes in regards to chronic diseases, including T2DM. This paper highlighted this model where a diabetes education programme/framework (provided by a team of board-certified family physicians, nurses, pharmacists, diabetes educators, and other supportive staff) was offered to patients attending a diabetes clinic. The education programme was in addition to any medication that the patient was already on.
The results observed in this study are impressive, as patients were engaged in an average of only 10 hours of the diabetes education programme, but generally showed optimized glycaemic and other cardiovascular risk control factors. Some of these patients have been suffering from diabetes for 5 to 10 years!
In my opinion, one of the most sobering facts in this study was the relatively ‘low level’ of education of the participants, of which nearly 90% were female. Of those approximately 60% were unable to read, with about 80% unable to count, and about 40% who were unable to do either! I therefore commend this group on the work that they are doing in trying to educate their patients. However, it seems that the problem of chronic disease is likely to continue, and may be bigger than what a research group could solve, if issues of education for women are not also addressed.
There are some major shortfalls in this study, most significantly the lack of a control group (who were not part of the education programme). However, these were addressed in the section on ‘Strengths and Limitations’ and therefore there is still good value in this work.
I can therefore recommend this article for publication with a few suggestions/minor comments that I outline below:
Line 58: “The prevalence of type 2 DM rose 57 from 23.7% in 2000 to 25.4% in 2009”. It is suggested to use more recent statistics or remove the statement.
Line 69: “Saudi population showed their preference to receive detailed information….” Reword for accuracy.
Line 128: The reader may be interested in knowing what ‘skills of problem solving’ and ‘healthy coping for better quality of life’ entails (without having to read the reference). Just a couple of lines should be sufficient.
Line 174: “Patients who did not reach the desired clinical 174 or biological outcomes…” It is worth clarifying at what stage of the study this occurred.
Figure 4: I found this confusing to understand, maybe another reviewer might find it easier. But if not, I wonder if there may be another way of displaying this figure, or better explaining the data on it.
Is there a hypothesis of what is expected beyond the six months duration of this study? Considering that the six months ended in July 2021, was there any follow up with these patients to know if the knowledge and practise that benefitted them by optimising glycaemic and other cardiovascular risk control factors, was sustainable?
Author Response
Reviewer 5
Reviewers General Comments
Thank you for giving me the opportunity to review this interesting and original article entitled “Impact of Patient-Centered and Self-Care Education on Diabetes Control in a Family Practice Setting”. The topic of this article is of great significance as it addresses the issue of an education gap around diabetes control. The authors rightfully highlighted that diabetes (mainly type II, T2DM) is of global concern, with an expected 700 million people who might be affected by the year 2045! Thus, there is an urgent need to implement effective care management of T2DM. There has been growing evidence that empowering patients to be more involved in their own health care contributes to better outcomes in regards to chronic diseases, including T2DM. This paper highlighted this model where a diabetes education programme/framework (provided by a team of board-certified family physicians, nurses, pharmacists, diabetes educators, and other supportive staff) was offered to patients attending a diabetes clinic. The education programme was in addition to any medication that the patient was already on.
The results observed in this study are impressive, as patients were engaged in an average of only 10 hours of the diabetes education programme, but generally showed optimized glycaemic and other cardiovascular risk control factors. Some of these patients have been suffering from diabetes for 5 to 10 years!
In my opinion, one of the most sobering facts in this study was the relatively ‘low level’ of education of the participants, of which nearly 90% were female. Of those approximately 60% were unable to read, with about 80% unable to count, and about 40% who were unable to do either! I therefore commend this group on the work that they are doing in trying to educate their patients. However, it seems that the problem of chronic disease is likely to continue, and may be bigger than what a research group could solve, if issues of education for women are not also addressed.
In Saudi Arabia, formal female education was delayed until 1959, when a Royal Decree was issued establishing a department to oversee the education of girls called the General Presidency for Girls Education, with branches in various parts of the country. The majority of patients in our study were females of old age.
There are some major shortfalls in this study, most significantly the lack of a control group (who were not part of the education programme). However, these were addressed in the section on ‘Strengths and Limitations’ and therefore there is still good value in this work.
Thank you.
I can therefore recommend this article for publication with a few suggestions/minor comments that I outline below:
Line 58: “The prevalence of type 2 DM rose 57 from 23.7% in 2000 to 25.4% in 2009”. It is suggested to use more recent statistics or remove the statement.
More recent data were utilized. [lines 56-57]
Line 69: “Saudi population showed their preference to receive detailed information….” Reword for accuracy.
Rephrase as follow:
“Patients in the local setting of Saudi Arabia showed their willingness to receive more information about their medical illness that includes its nature, progression, and available management options” [lines 66-68]
Line 128: The reader may be interested in knowing what ‘skills of problem solving’ and ‘healthy coping for better quality of life’ entails (without having to read the reference). Just a couple of lines should be sufficient.
This paragraph was added to elaborate on these points:
“Problem-solving skills are highly needed for patients with diabetes to enable them to handle acute and chronic complications of the diseases or its medications such as hypoglycemia and hyperglycemia. As some of these problems might be life-threatening, a systematic approach is taught to patients during these sessions for early anticipation and prompt intervention. Health coping strategies include teaching patients strategies to fulfill health care and psychosocial obligations for diabetes care as healthy diet alternative, expressing emotions, and incorporating physical activities in the daily life of patients” [lines 125-132].
Line 174: “Patients who did not reach the desired clinical 174 or biological outcomes…” It is worth clarifying at what stage of the study this occurred.
This was clarified
This statement was added to the text: ” at three months and six months of the beginning of the study” [line 175]
Figure 4: I found this confusing to understand, maybe another reviewer might find it easier. But if not, I wonder if there may be another way of displaying this figure, or better explaining the data on it.
Agree. Figure 4 was reformatted to be more straightforward to understand.
Is there a hypothesis of what is expected beyond the six months duration of this study? Considering that the six months ended in July 2021, was there any follow up with these patients to know if the knowledge and practise that benefitted them by optimising glycaemic and other cardiovascular risk control factors, was sustainable?
Agree. However, the study was IRB approved for only six months of follow-up, and we felt we had to comply with this approval.
Round 2
Reviewer 2 Report
Thank you for addressing my comments.
Author Response
Much appreciated.
Reviewer 4 Report
I can't find the answers to my questions. The answer to all of my questions was
"All these points and queries were responded to in the data analysis section with a more detailed description of the statistical tests used and the justifications for their use. [lines 219-228, 276, 291-292]"
This is not the way to answer all of my comments. Please provide a report to answer each question in my report.
Author Response
First of all, I appreciate you would kindly accept my apology for not responding to your valuable queries one by one. I thought all questions were responded to in the revised version and highlighted in RED. Below are the responses to these queries, one by one.
Reviewer 4
In general, the contribution from this paper in its current form is very marginal, thus not significant and sufficient to be considered as a full length research paper.
The Data Analysis (subsection 3.3) and Results (Section 4) need significant improvements. The statistical criteria, techniques and discussions that are used in this study are very trivial. These basic statistical measures are used without justifications. Therefore, more advanced techniques are needed. For example,
- Why the interquartile range (IQR) is used in your study?
The median and interquartile range were used in place of the mean and standard deviation, as the data/parameters were not normally distributed and had high extreme high and low extreme values. Moreover, the sample size was small.
- Is 95% Confidence Interval better than the IQR for your study?
First, the interquartile range (IQR) is the difference between the third and first quartiles, which is a single number, not an interval. However, IQR is better in this study when the data are not normally distributed.
- A comparison study between these two statistical measures is needed.
In a perfectly normal distribution (z), 1.5 IQR corresponds to the 16th and 84th percentiles, which is a 69% confidence interval. However, the distribution of data was not normal. Thus, we assumed the median and IQR might be sufficient and justifiable in this study.
- What do you mean by Median (IQR)? Do you mean the second quartile Q2?
The median is the second quartile, and the interquartile range (IQR) is the difference between the third and first quartiles.
- Why "χ2" test is used in your study? Some discussions are needed.
Thank you. Cochran’s Q test for related samples, a non-parametric statistical test, was applied to test the change of prevalence of different clinical and laboratory parameters three and six months after education intervention. Its value is presented as a chi-square value, which is different from the Pearson chi-square.
Note: The chi-square values were removed from the abstract (while the p-values were left) to avoid the reader's confusion between Cochran’s Q test and Pearson's chi-square.
- Since you have parameters before and after education, can paired samples t-test be used? Some discussions are needed.
The Friedman test, a non-parametric statistical test similar to the parametric repeated measures ANOVA, was used to detect differences in the impact of intervention across multiple test attempts. This test allows for the comparison between every two paired sets of data. Paired samples t-test was not used because the data were not normally distributed and the sample size was small.
- The figures and their related discussions are very trivial. I suggest the authors to use more advanced statistical techniques and discussions.
All these points and queries were responded to in the data analysis section, with a more detailed description of the statistical tests used and the justifications for their use. [lines 219-228, 276, 291-292]
I hope these responses will be satisfactory, and we will be more than happy to answer any more queries.